# PHOTONAI—A Python API for rapid machine learning model development

**Ramona Leenings**[1,2]* , **Nils Ralf Winter**[1] , **Lucas Plagwitz**[1] , **Vincent Holstein**[1] ,
**Jan Ernsting**[1,2] , **Kelvin Sarink**[1] , **Lukas Fisch**[1] , **Jakob Steenweg**[1] , **Leon Kleine-
Vennekate**[1] , **Julian Gebker**[1] , **Daniel Emden**[1] , **Dominik Grotegerd**[1] , **Nils Opel**[1] ,
**Benjamin Risse**[2] , **Xiaoyi Jiang**[2] , **Udo Dannlowski**[1] , **Tim Hahn**[1]

**1** Institute for Translational Psychiatry, University of Münster, Münster, Germany, **2** Faculty of Mathematics and Computer Science, University of Münster, Münster, Germany

☯ These authors contributed equally to this work.
* leenings@uni-muenster.de

**Data Availability Statement:** https://doi.org/10. 1186/s12911-020-1023-5 Davide Chicco, Giuseppe Jurman: Machine learning can predict survival of patients with heart failure from serum

## Abstract

PHOTONAI is a high-level Python API designed to simplify and accelerate machine learning model development. It functions as a unifying framework allowing the user to easily access and combine algorithms from different toolboxes into custom algorithm sequences. It is especially designed to support the iterative model development process and automates the repetitive training, hyperparameter optimization and evaluation tasks. Importantly, the work-flow ensures unbiased performance estimates while still allowing the user to fully customize the machine learning analysis. PHOTONAI extends existing solutions with a novel pipeline implementation supporting more complex data streams, feature combinations, and algorithm selection. Metrics and results can be conveniently visualized using the PHOTONAI Explorer and predictive models are shareable in a standardized format for further external validation or application. A growing add-on ecosystem allows researchers to offer data modality specific algorithms to the community and enhance machine learning in the areas of the life sciences. Its practical utility is demonstrated on an exemplary medical machine learning problem, achieving a state-of-the-art solution in few lines of code. Source code is publicly available on Github, while examples and documentation can be found at www. photon-ai.com.

## Introduction

In recent years, the interest in machine learning for medical, biological, and life science research has significantly increased. Technological advances develop with breathtaking speed. The basic workflow to construct, optimize and evaluate a machine learning model, however, has remained virtually unchanged. In essence, it can be framed as the (systematic) search for the best combination of data processing steps, learning algorithms, and hyperparameter values under the premise of unbiased performance estimation.

Subject to the iteratively optimized workflow is a machine learning pipeline, which in this context is defined as the sequence of algorithms subsequently applied to the data. To

creatinine and ejection fraction alone. BMC Medical Informatics and Decision Making 20, 16 (2020) https://www.kaggle.com/andrewmvd/heart-failure-clinical-data.

**Funding:** This work was supported by grants from the Interdisciplinary Center for Clinical Research (IZKF, https://www.medizin.uni-muenster.de/izkf.html) of the medical faculty of Münster (grant MzH 3/020/20 to TH and grant Dan3/012/17 to UD) and the German Research Foundation (DFG, https://www.dfg.de/, grants HA7070/2-2, HA7070/3, HA7070/4 to TH). The funders had no role in study design, data collection and analysis, decision to publish, or preparation of the manuscript.

**Competing interests:** The authors have declared that no competing interests exist.

begin with, the data is commonly prepared by successively applying several processing steps such as normalization, imputation, feature selection, dimensionality reduction, data augmentation, and others. The altered data is then forwarded to one or more learning algorithms which internally derive the best fit for the learning task and finally yield predictions.

In practice, researchers select suitable preprocessing and learning algorithms from different toolboxes, learn toolbox-specific syntaxes, decide for a training and testing scheme, manage the data flow and, over time, iteratively optimize their choices. Importantly, all of this is done while preventing data leakage, calculating performance metrics, adhering to (nested) cross-validation best practices, and searching for the optimal (hyperparameter-) configuration.

A multitude of high-quality and well-maintained open-source toolboxes offer specialized solutions, each for a particular subdomain of machine learning-related (optimization) problems.

## Existing solutions: Specialized open-source toolboxes

In the field of (deep) neural networks, libraries such as *Tensorflow*, *Theano*, *Caffe* and *PyTorch* [1–4] offer domain-specific implementations for nodes, layers, optimizers, as well as evaluation and utility functions. On top of that, higher level Application Programming Interfaces (APIs) such as *Keras* and *fastai* [5, 6] offer expressive syntaxes for accelerated and enhanced development of deep neural network architectures.

In the same manner, the *scikit-learn* [7] toolbox, has evolved as one of the major resources of the field, covering a very broad range of regression, classification, clustering, and preprocessing algorithms. It has established the de-facto standard interface for data processing and learning algorithms, and, in addition, offers a wide range of utility functions, such as cross-validation schemes and model evaluation metrics.

Next to these general frameworks, other libraries in the software landscape offer functionalities to address more specialized problems. Prominent examples are the *imbalanced-learn* toolbox [8], which provides numerous over- or under-sampling methods, or modality-specific libraries such as *nilearn* and *nibabel* [9, 10] which offer utility functions for accessing and preparing neuroimaging data.

On top of that, the software landscape is complemented by several hyperparameter optimization packages, each implementing a different strategy to find the most effective hyperparameter combination. Next to Bayesian approaches, such as *Scikit-optimize* or *SMAC* [11, 12], there are packages implementing evolutionary strategies [13] or packages approximating gradient descent within the hyperparameter space [14, 15]. Each package requires specific syntax and unique hyperparameter space definitions.

Finally, there are approaches uniting all these components into algorithms that automatically derive the best model architecture and hyperparameter settings for a given dataset. Libraries such as *auto-sklearn*, *TPOT*, *AutoWeka*, *Auto-keras*, *AutoML*, *Auto-Gluon* and others optimize a specific set of data-processing methods, learning algorithms and their respective hyperparameters [16–22]. While very intriguing, these libraries aim at full automation—neglecting the need for customization and foregoing the opportunity to incorporate high-level domain knowledge in the model architecture search. Especially the complex and often high-dimensional data structure native to medical and biological research requires the integration and application of modality-specific processing and often entails the development of novel algorithms.

## Current shortcoming: Manual integration of cross-toolbox algorithm sequences

Currently, iterative model development approaches across different toolboxes as well as design and optimization of custom algorithm sequences are barely supported. For a start, *scikit-learn* has introduced the concept of pipelines, which successively apply a list of processing methods (referred to as transformers) and a final learning algorithm (called estimator) to the data. The pipeline directs the data from one algorithm to another and can be trained and evaluated in (simple) cross-validation schemes, thereby significantly reducing programmatic overhead. Scikit-learn's consistent usage of standard interfaces enables the pipeline to be subject to scikit-learn's inherent hyperparameter optimization strategies based on random- and grid-search. While being a simple and effective tool, several limitations still remain. For one, hyperparameter optimization requires a nested cross-validation scheme, which is not inherently enforced. Second, a standardized solution for easy integration of custom or third-party algorithms is not considered. In addition, several repetitive tasks, such as metric calculations, logging, and visualization lack automation and still need to be handled manually. Finally, the pipeline can not handle adjustments to the target vector, thereby excluding algorithms for e.g. data augmentation or handling class imbalance.

## Major contributions of PHOTONAI: Supporting a convenient development workflow

To address these issues, we propose *PHOTONAI* as a high-level Python API that acts as a mediator between different toolboxes. Established solutions are conveniently accessible or can be easily added. It combines an automated supervised machine learning workflow with the concept of custom machine learning pipelines. Thereby it is able to considerably accelerate design iterations and simplify the evaluation of novel analysis pipelines. In essence, *PHOTONAI*'s major contributions are:

**Increased accessibility.** By pre-registering data processing methods, learning algorithms, hyperparameter optimization strategies, performance metrics, and other functionalities, the user can effortlessly access established machine learning implementations via simple keywords. In addition, by relying on the established scikit-learn object API [23], users can easily integrate any third-party or custom algorithm implementation.

**Extended pipeline functionality.** A simple to use class structure allows the user to arrange selected algorithms into single or parallel pipeline sequences. Extending the pipeline concept of scikit-learn [7], we add novel functionality such as flexible positioning of learning algorithms, target vector manipulations, callback functions, specialized caching, parallel data-streams, Or-Operations, and other features as described below.

**Automation.** PHOTONAI can automatically train, (hyperparameter-) optimize and evaluate any custom pipeline. Importantly, the user designs the training and testing procedure by selecting (nested) cross-validation schemes, hyperparameter optimization strategies, and performance metrics from a range of pre-integrated or custom-built options. Thereby, development time is significantly decreased and conceptual errors such as information leakage between training, validation, and test set are avoided. Training information, baseline performances, hyperparameter optimization progress, and test performance evaluations are persisted and can be visualized via an interactive, browser-based graphical interface (PHOTONAI Explorer) to facilitate model insight.

**Model sharing.** A standardized format for saving, loading, and distributing optimized and trained pipeline architectures enables model sharing and external model validation even for non-expert users.

## Materials and methods

In the following, we will describe the automated supervised machine learning workflow implemented in PHOTONAI. Subsequently, we will outline the class structure, which is the core of its expressive syntax. At the same time, we will highlight its current functionalities, and finally, provide a hands-on example to introduce PHOTONAI's usage. Lastly, we close with discussing current challenges and future developments.

### Software architecture and workflow

PHOTONAI automatizes the supervised machine learning workflow according to user-defined parameters (see pseudocode in Listing 1). In a nutshell, cross-validation folds are derived to iteratively train and evaluate a machine learning pipeline following the hyperparameter optimization strategy's current parameter value suggestions. Performance metrics are calculated, the progress is logged and finally, the best hyperparameter configuration is selected to train a final model. The training, testing, and optimization workflow is automated, however, it is important to note that it is parameterized by user choices and therefore fully customized.

In order to achieve an efficient and expressive customization syntax, PHOTONAI's class architecture captures all workflow- and pipeline-related parameters into distinct and combinable components (see Fig 1). A central management class called *Hyperpipe*—short for hyperparameter optimization pipeline—handles the setup of the pipeline and executes the training and test procedure according to user choices. Basis to the data flow is a custom *Pipeline*

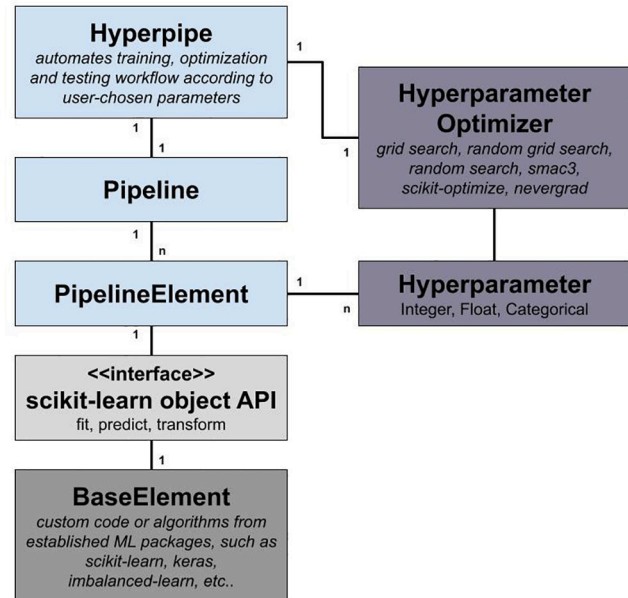

**Fig 1. Class architecture.** The PHOTONAI framework is built to accelerate and simplify the design of machine learning models. It adds an abstraction layer to existing solutions and is thereby able to simplify, structure, and automate the training, optimization, and testing workflow. Importantly, the pipeline and the workflow are subject to user choices as the user selects a sequence of processing and learning algorithms and parameterizes the optimization and validation workflow. The here depicted class diagram shows PHOTONAI's core structure. The central element is the *Hyperpipe* class, short for hyperparameter optimization pipeline, which manages a pipeline and the associated training, optimization, and testing workflow. The *Pipeline* streams data through a sequence of *n PipelineElements*. PHOTONAI relies on the established scikit-learn [7] object API, to integrate established or custom algorithms (*BaseElements*) into the workflow. *PipelineElements* can have *n* hyperparameters which are subject to optimization by a hyperparameter optimization strategy.

implementation, which streams data through a sequence of *PipelineElement* objects, the latter of which represent either established or custom algorithm implementations. In addition, clear interfaces and several utility classes allow the integration of custom solutions, adjust the training and test procedure and build parallel data streams. In the following, PHOTONAI's core classes and their respective features will be further detailed.

**Core framework—The hyperpipe class.** PHOTONAI's core functionality is encapsulated in a class called *Hyperpipe*, which controls all workflow and pipeline-related parameters and manages the cross-validated training and testing procedure (see Listing 1). In particular, it partitions the data according to the cross-validation splits, requests hyperparameter configurations, trains and evaluates the pipeline with the given configuration, calculates performance metrics, and coordinates the logging of all results and metadata such as e.g. computation time. In addition, the *Hyperpipe* ranks all tested hyperparameter configurations based on a user-selected performance metric and yields a final (optimal) model trained with the best performing hyperparameter configuration. Further, a baseline performance is established by applying a simple heuristic [24]. This aids in assessing model performance and facilitates interpretation of the results.

**Algorithm 1** Pseudocode for PHOTONAI's training, hyperparameter optimization and testing workflow as implements in the Hyperpipe class

```
Input:
(1) Pipeline, sequence of algorithms, pipeline
(2) Performance metrics, metrics
(3) Hyperparameter Optimization Strategy, hpo
(4) Outer Cross-Validation Strategy, ocv
(5) Inner Cross-Validation Strategy, icv
(6) Features X and Targets y, data
(7) Performance Expectations, performance_constraints
```

```
1  for outer_fold = 1, 2, ... T ∈ ocv.split(data) do
2    outer_fold_data = data[outer_fold_T]
3    dummy_performance = apply_dummy_heuristic(outer_fold_data)
4
5    hpo.initialize_hyperparameter_space(pipeline)
6    for hp_config in hpo.ask() do
7      for inner_fold = 1, 2, ..V ∈ icv.split(outer_fold_data) do
8        inner_data = outer_fold_data[inner_fold_V]
9        hp_performance = train_and_test(pipeline, hp_config,
10                         metrics, inner_data)
11       hpo.tell(hp_performance)
12       if performance_constraints then
13         if hp_performance < performance_constraints then
14           break
15         end if
16       end if
17      end for
18      val_performance = mean([hp_performance_1, ..., hp_performance_V])
19      if hp_config_performance < best_performance then
20        best_outer_fold_configT = hp_config
21      end if
22    end for
23    test_performance = train_and_test(pipeline,
                                        best_outer_fold_config,
24                                      metrics, outer_fold_data)
25  end for
26  overall_best_performance = argmax([test_performance_1, ...,
    test_performance_T])
```

```
27 overall_best_config = [best_outer_configs1, ...,
28                best_outer_configT][overall_best_performance]
29 pipeline.set_params(overall_best_config)
30 pipeline.fit(X, y)
31 pipeline.save()
```

**Listing 1**. Setting the parameters to control the training, hyperparameter optimization and testing workflow using the *Hyperpipe* class.

```
1 pipe = Hyperpipe('example_project',
2                optimizer='sk_opt',
3                optimizer_params={'n_configurations': 25},
4                metrics=['accuracy', 'precision', 'recall'],
5                best_config_metric='accuracy',
6                outer_cv=KFold(n_splits=3),
7                inner_cv=KFold(n_splits=3))
```

**The PHOTONAI pipeline—Extended pipeline features.** The *Hyperpipe* relies on a custom pipeline implementation that is conceptually related to the scikit-learn pipeline [25] but extends it with four core features. First, it enables the positioning of learning algorithms at an arbitrary position within the pipeline. In case a *PipelineElement* is identified that a) provides no *transform* method and b) yet is followed by one or more other *PipelineElements*, it automatically calls *predict* and delivers the output to the subsequent pipeline elements. Thereby, learning algorithms can be joined to ensembles, used within sub pipelines, or be part of other custom pipeline architectures without interrupting the data stream.

Second, it allows for a dynamic transformation of the target vector anywhere within the data stream. Common use-cases for this scenario include data augmentation approaches—in which the number of training samples is increased by applying transformations (e.g. rotations to an image)—or strategies for an imbalanced dataset, in which the number of samples per class is equalized via e.g. under- or oversampling.

Third, numerous use-cases rely on data not contained in the feature matrix at runtime, e.g. when aiming to control for the effect of covariates. In PHOTONAI, additional data can be streamed through the pipeline and is accessible for all pipeline steps while—importantly—being matched to the (nested) cross-validation splits.

Finally, PHOTONAI implements pipeline callbacks which allow for live inspection of the data flowing through the pipeline at runtime. *Callbacks* act as pipeline elements and can be inserted at any point within the pipeline. They must define a function delegate which is called with the same data that the next pipeline step will receive. Thereby, a developer may inspect e.g. the shape and values of the feature matrix after a sequence of transformations has been applied. Return values from the delegate functions are ignored so that after returning from the delegate call, the original data is directly passed to the next processing step.

**Listing 2**. Algorithms can be accessed via keywords and are represented together with all potential hyperparameter values.

```
1 # add two preprocessing algorithms to the data stream
2 pipe += PipelineElement('PCA',
3                hyperparameters={'n_components':
4                        FloatRange(0.5, 0.8, step = 0.1)},
5                test_disabled = True)
6
7 pipe += PipelineElement('ImbalancedDataTransformer',
8                hyperparameters={'method_name':
9                        ['RandomUnderSampler','SMOTE']},
10                test_disabled = True)
```

**The pipeline element—Conveniently access cross-toolbox algorithms.** In order to integrate a particular algorithm into the pipeline's data stream, PHOTONAI implements the *PipelineElement* class. This can either be a data processing algorithm, in reference to the *scikit-learn* interface also called transformer, or a learning algorithm, also referred to as estimator. By selecting and arranging *PipelineElements*, the user designs the ML pipeline. To facilitate this process, it enables convenient access to various established implementations from state-of-the-art machine learning toolboxes: With an internal registration system that instantiates class objects from a keyword, import, access, and setup of different algorithms is significantly simplified (see Listing 2). Relying on the established scikit-learn object API [23], users can integrate any third-party or custom algorithm implementation. Once registered, custom code fully integrates with all PHOTONAI functionalities thus being compatible with all other algorithms, hyperparameter optimization strategies, PHOTONAI's pipeline functionality, nested cross-validation, and model persistence.

## Hyperparameter optimization strategies

Hyperparameters directly control the behavior of algorithms and may have a substantial impact on model performance. Therefore, unlike classic hyperparameter optimization, PHOTONAI's hyperparameter optimization encompasses the hyperparameters of the entire pipeline—not only the learning algorithm's hyperparameters as is usually done. The *PipelineElement* provides an expressive syntax for the specification of hyperparameters and their respective value ranges (see Listing 2). In addition, PHOTONAI conceptually extends the hyperparameter search by adding an on and off switch (a parameter called *test_disabled*) to each *PipelineElement*, allowing the hyperparameter optimization strategy to check if skipping an algorithm improves model performance. Representing algorithms together with their hyperparameter settings enables seamless switching between different hyperparameter optimization strategies, ranging from (random) grid search to more advanced approaches such as Bayesian or evolutionary optimization [11–13] Custom hyperparameter optimization strategies can be integrated via an extended an ask- and tell-interface or by accepting an objective function defined by PHOTONAI.

## Parallel data streaming

**The *Switch* element—Optimizing algorithm selection.** Building ML pipelines involves comparing different pipelines with each other. While in most state-of-the-art ML toolboxes the user has to define and benchmark each pipeline manually, in PHOTONAI it is possible to evaluate several possibilities at once. Specifically, the *Switch* object is interchanging several algorithms at the same pipeline position, representing an OR-Operation (see Fig 2). With data processing steps, learning algorithms and their hyperparameters intimately entangled, this enables algorithm selection to be part of the hyperparameter optimization process. For an example usage of the *Switch* element see example code on github.

**The *Stack* element—Combining data streams.** The *Stack* object acts as an AND-Operation. It allows several algorithms to share a particular pipeline position, streams the data to each element and horizontally concatenates the respective outputs (see Fig 2 and Listing 3 or demo code on github). Thus, new feature matrices can be created by processing the input in different ways and likewise, ensembles can be built by training several learning algorithms in parallel.

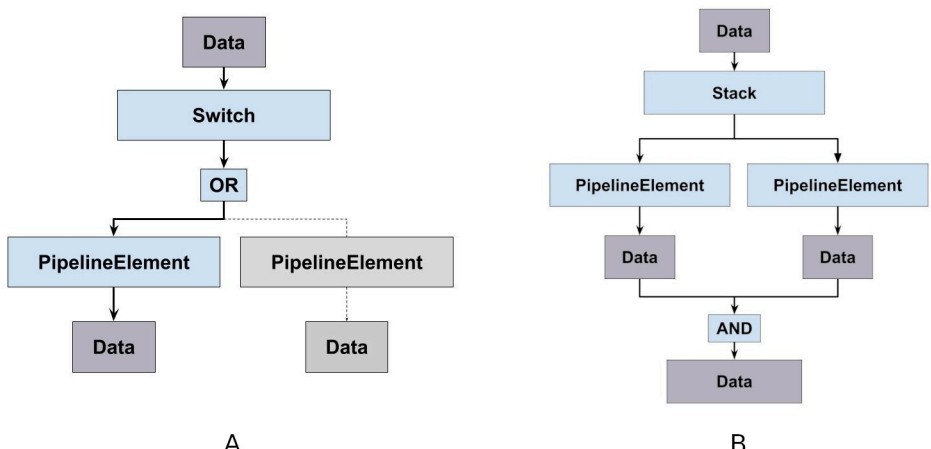

**Fig 2. Parallel pipeline elements. A**: The *Switch* class represents an OR-Operation and can be placed anywhere in the sequence to interchange and compare different algorithms at the same pipeline position. **B**: The *Stack* represents an AND-Operation and contains several algorithms to share a particular pipeline position. It streams the data to each element and horizontally concatenates the respective outputs (see Listing 3). Next to generating new feature matrices through several processing steps at runtime or building classifier ensembles, it can, in addition, be used in combination with the branch element.

**Listing 3**. Using the *Stack* object, two learning algorithms can be trained in parallel resulting in various predictions that can e.g. to be fed into a subsequent meta-learner to create an ensemble.

```
1
2 # set up two learning algorithms in an ensemble
3 ensemble = Stack('estimators', use_probabilities = True)
4 ensemble += PipelineElement('DecisionTreeClassifier',
5                     criterion='gini',
6                     hyperparameters={'min_samples_split':
7                             IntegerRange(2, 4)})
8 ensemble += PipelineElement('LinearSVC',
9                     hyperparameters={'C': FloatRange(0.5, 25)})
10
11 pipe += ensemble
```

**The *Branch* element—Building nested pipelines.** Finally, the *Branch* class constitutes a parallel sub-pipeline containing a distinct sequence of PipelineElements. It can be used in combination with the *Switch* and *Stack* elements enabling the creation of complex pipeline architectures integrating parallel sub-pipelines in the data flow (see usage example on github). This could be particularly useful when deriving distinct predictions from several data subdomains, such as different brain regions, and further apply a voting strategy to the respective outputs.

## Increasing workflow efficiency

**Accelerated computation.** Several computational shortcuts are implemented in order to most efficiently use available resources. PHOTONAI allows specifying lower or upper bounds which the performance of a hyperparameter configuration has to exceed. Only then, the configuration is further evaluated in the remaining cross-validation folds, thereby accelerating hyperparameter search [16]. In addition, PHOTONAI can compute outer cross-validation folds in parallel relying on the python library *dask* [26]. It is compatible with any custom parallelized model implementation, e.g. for training a multi GPU model. Finally, PHOTONAI is able to reuse data already calculated: It implements a caching strategy that is specifically

adapted to handle the varying datasets evolving from the nested cross-validation data splits as well as partially overlapping hyperparameter configurations.

**Model distribution.** After identifying the optimal hyperparameter configuration, the *Hyperpipe* trains the pipeline with the best configuration on all available data. The resulting model including all transformers and estimators is persisted as a single file in a standardized format, suffixed with '.*photon*'. It can be reloaded to make predictions on new, unseen data. The .photon format facilitates model distribution, which is crucial for external model validation and thus at the heart of ML best practice, we also created a dedicated online model repository to which users can upload their models to make them publicly available. If the model is persisted in the .photon-format, others can download the file and make predictions without extensive system setups or the need to share data.

**Logging and visualization.** PHOTONAI provides extensive result logging including both performances and metadata generated through the hyperparameter optimization process. Each hyperparameter configuration tested is archived including all performance metrics and complementary information such as computation time and the training, validation, and test indices.

Finally, all results can be visualized by uploading the JSON output file to a JavaScript web application called Explorer. It provides a visualization of the pipeline architecture, analysis design, and performance metrics. Confusion matrices (for classification problems) and scatter plots (for regression analyses) with interactive per-fold visualization of true and predicted values are shown. All evaluated hyperparameter configurations can be sorted and are searchable. In addition, the course of the hyperparameter optimization strategy over time is visualized (see Fig 3).

## Example usage

In the following, we will provide a hands-on example for using PHOTONAI to predict heart failure from medical data. To run the example, download the data available on kaggle [27] and install PHOTONAI either by cloning it from Github or installing it via pip using:

```
1 pip install photonai
```

The complete example code can be downloaded from Github using this link.

### Heart failure data

In the following, we will develop a model to predict mortality in the context of heart failure based on medical records [27, 28]. The dataset consists of data from 299 patients (105 female, 194 male) in the age between 40 and 95. It provides 13 features per subject: age and gender, several clinical blood markers, information about body functions as well as the presence of comorbidities (anemia, diabetes), and lifestyle impacts (smoking). Finally, a boolean value indicates whether a subject died during the follow-up period, which spans 4 to 285 days. In approximately 68 percent of cases the patients survived while approximately 32 percent of the patients die due to heart failure.

### Hyperpipe setup

First, we will define the training, optimization, and evaluation workflow in an initial call to the *Hyperpipe* class. The python code in Listing 4 shows the PHOTONAI code defining both the data flow and the training and test procedure. After importing the relevant packages and loading the data, we instantiate a *Hyperpipe* and choose the workflow parameters as follows:

- For the outer cross-validation loop, we specify 100 shuffled iterations each holding out a test set of 20 percent. For the inner cross-validation loop, we select a ten-fold cross-validation. (lines 13-14).

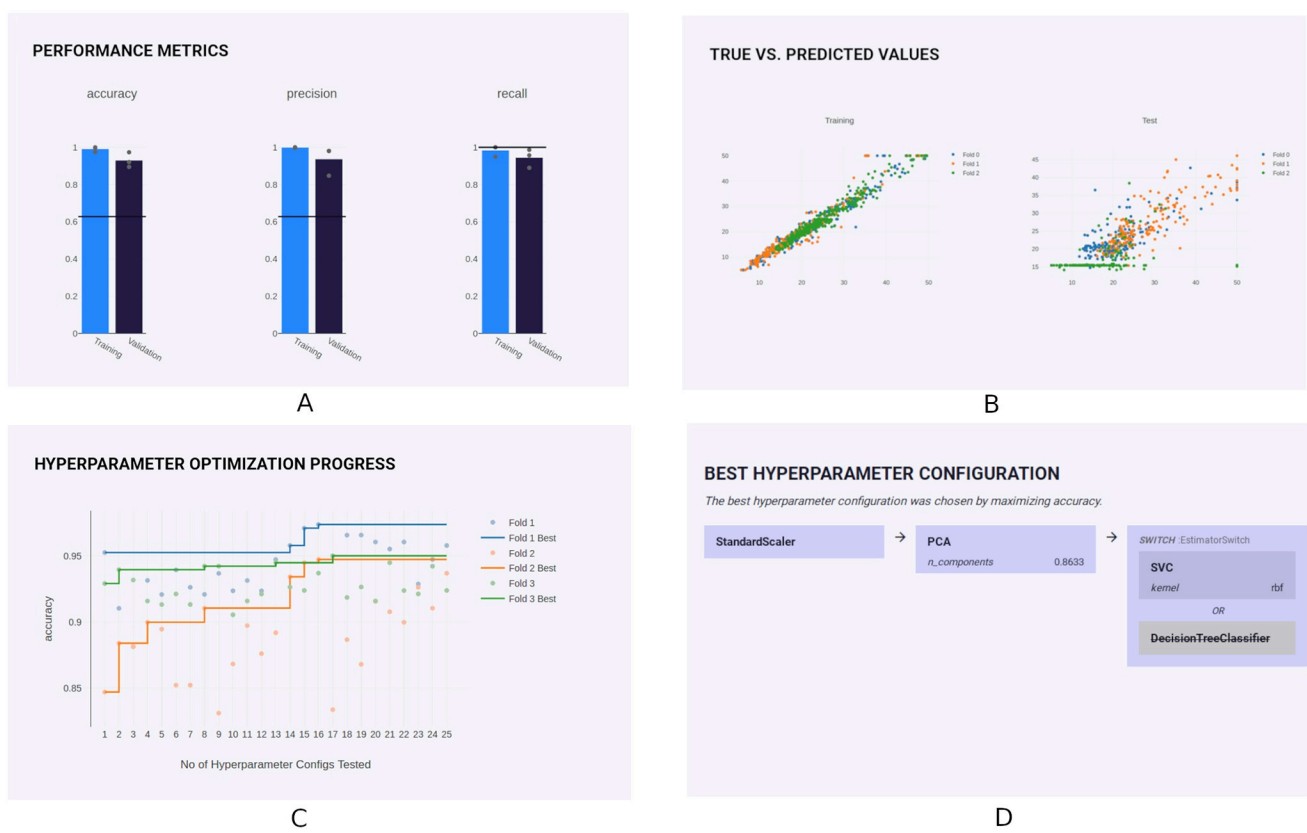

**Fig 3. PHOTONAI explorer.** Example plots of PHOTONAI's result visualization tool called Explorer. **A**: User-defined performance metrics, here accuracy, precision and recall, for both training (blue) and test (dark) set. The horizontal line indicates a baseline performance stemming from a simple heuristic. **B**: For regression problems, true and predicted values are visualized in a scatter plot on both train (left) and test (right) set. The values are generated by the best model found in each outer folds, respectively. **C**: Hyperparameter optimization progress is depicted over time for each outer fold. **D**: Pipeline elements and their arrangement is visualized including the best hyperparameter value of each item.

- To measure model performance, we specify that f1 score, Matthews correlation coefficient, balanced accuracy, as well as sensitivity and specificity are to be calculated (lines 16-17).

- We optimize the pipeline for f1 score, as it maximizes both sensitive and specific predictions, which is particularly important in medical contexts. (line 18).

- To save computational resources and time, we enable caching by specifying a cache folder (line 22). This is particularly useful in examples where there are a lot of partially overlapping hyperparameters to be tested.

- Finally, we specify a folder to which the output is written (line 21) and set the verbosity of the console log to 1. At this verbosity level, information on every tested hyperparameter configuration and its respective performance estimate is printed to the console.

   After the hyperpipe has been defined, we can design the flow of the data by adding algorithms and respective hyperparameters to the pipeline.

- First, data is normalized using scikit-learn's StandardScaler which both centers the data and scales it to unit variance (line 26).

- Second, we impute missing values with the mean values per feature of the training set by calling scikit-learn's SimpleImputer (line 27).

Of note, we consider use cases 1 to 3 (see below) to be exploratory analyses. We believe this simulates a naturalistic workflow of machine learning projects where different algorithms, feature preprocessing and hyperparameters are tested in a manual fashion. However, if done incorrectly, this inevitably leads to a manual over-fitting to the data at hand, which is especially troublesome in high-stake medical problems with small datasets. In this context, manual over-fitting happens implicitly when data scientists optimize algorithms and hyperparameters by repeatedly looking at cross-validated test performance. In PHOTONAI, this problem can easily be avoided by setting the *Hyperpipe* parameter *use_test_set* to *False*. This way, PHOTONAI will still apply nested cv but will only report validation performances from the inner cv loop, not the outer cv test data. In the final use case 4, *use_test_set* is set to *True* to estimate final model performance and generalizability on the actual test sets.

## Use case 1—Estimator selection

Although some rules of thumb for selecting the correct algorithm do exist, knowing the optimal learning algorithm for a specific task a priori is impossible (no free lunch theorems [29]). Therefore, the possibility to automatically evaluate multiple algorithms within nested cross-validation is crucial to efficient and unbiased machine learning analyses. In this example, we first train a machine learning pipeline and consider three different learning algorithms that we find appropriate for this learning problem. These algorithms are added to the PHOTONAI *Hyperpipe* in addition to the scaling and imputing preprocessing steps defined above.

**Setup.**

- To compare different learning algorithms, an Or-Element called Switch is added to the pipeline that toggles between several learning algorithms (see Listing 5). Here, we compare a random forest (RF), gradient boosting (GB), and a support vector machine (SVM) against each other. Again, all algorithms are imported from scikit-learn, and for every element we specify algorithm-specific hyperparameters that are automatically optimized.

- To efficiently optimize hyperparameters of different learning algorithms, the switch optimizer in PHOTONAI can be used which optimizes each learning algorithm in an individual hyperparameter space (line 19 in Listing 4). We apply Bayesian optimization to each space respectively and limit the number of tested configurations to 10 (line 20 in Listing 4).

Finally, we can start the training, optimization, and test procedure by calling *Hyperpipe.fit()*. After the pipeline optimization has finished, we extract not only the overall best hyperparameter configuration and its respective performance, but also the best configuration performance per learning algorithm (RF, GB, SVM, see line 21 in Listing 5).

**Listing 4**. PHOTONAI code to define an initial training, optimization and test proecdure for the heart failure dataset. The pipeline normalizes the data and imputes missing values.

```
1 import pandas as pd
2 from sklearn.model_selection import KFold, ShuffleSplit
3 from photonai.base import Hyperpipe, PipelineElement, Switch
4 from photonai.optimization import FloatRange, IntegerRange,
MinimumPerformanceConstraint
5
6 # load data
7 df = pd.read_csv('./heart_failure_clinical_records_dataset.csv')
8 X = df.iloc[:, 0:12]
9 y = df.iloc[:, 12]
```

```
10
11 # setup training and test workflow
12 pipe = Hyperpipe('heart_failure',
13             outer_cv = ShuffleSplit(n_splits = 100,
               test_size = 0.2),
14             inner_cv = KFold(n_splits = 10, shuffle = True),
15             use_test_set = False,
16             metrics=['balanced_accuracy', 'f1_score',
                     'matthews_corrcoef',
17                 'sensitivity', 'specificity'],
18             best_config_metric='f1_score',
19             optimizer='switch',
20             optimizer_params={'name': 'sk_opt',
               'n_configurations': 10},
21             project_folder='./tmp',
22             cache_folder='./cache',
23             verbosity = 1)
24
25 # arrange a sequence of algorithms subsequently applied
26 pipe += PipelineElement('StandardScaler')
27 pipe += PipelineElement('SimpleImputer')
28
29 # learning algorithm's will be added here
30 ...
31 #
32
33 # start the training, optimization and test procedure
34 pipe.fit(X, y)
```

**Listing 5**. PHOTONAI code to define three learning algorithms that are tested by PHOTO-
NAI automatically through an OR-element, the PHOTONAI Switch.

```
1 # compare different learning algorithms in an OR_Element
2 estimators = Switch('estimator_selection')
3
4 estimators += PipelineElement('RandomForestClassifier',
5                     criterion='gini',
6                     bootstrap = True,
7                     hyperparameters={'min_samples_split':
                     IntegerRange(2, 30),
8                               'max_features': ['auto', 'sqrt',
                                 'log2']})
9
10 estimators += PipelineElement('GradientBoostingClassifier',
11                 hyperparameters={'loss': ['deviance',
                   'exponential'],
12                             'learning_rate': FloatRange
                               (0.001, 1,
13                             "logspace")})
14 estimators += PipelineElement('SVC',
15                 hyperparameters={'C': FloatRange(0.5, 25),
16                             'kernel': ['linear', 'rbf']})
17 pipe += estimators
18
19 pipe.fit(X, y)
20
21 pipe.results_handler.get_mean_of_best_validation_configs_per_esti-
mator()
```

**Table 1. Validation performance metrics for three different pipeline setups.**

| Pipeline | f1 | matthews corr | BACC | sens | spec |
|---|---|---|---|---|---|
| Estimator selection pipeline | 0.7504 | 0.6583 | 0.8217 | 0.9144 | 0.7289 |
| + Lasso feature selection | 0.7496 | 0.6570 | 0.8211 | 0.7300 | 0.9123 |
| + class balancing | 0.7644 | 0.6619 | 0.8384 | 0.8210 | 0.8557 |

*Notes*: matthews corr = Matthews correlation coefficient, BACC = balanced accuracy, sens = sensitivity, spec = specificity

**Results.** The results of the initial estimator selection analysis are given in the first line of Table 1. We observe an f1 score of 75% and a Matthews correlation coefficient of 65%. The best config found by the hyperparameter optimization strategy applied the Random Forest classifier, which thus in this case outperforms gradient boosting and the Support Vector Machine.

**Listing 6**. Code for adding a feature selection pipeline element that uses Lasso coefficients to rank and remove features.

```
1 pipe += PipelineElement('LassoFeatureSelection',
2                 hyperparameters={'percentile': FloatRange(0.1,
                                                            0.5),
3                                  'alpha': FloatRange(0.5, 5,
4                                   range_type="logspace")})
```

## Use case 2—Feature selection

Next, we will evaluate the effect of an additional feature selection step. This can be done, e.g., by analyzing a linear model's normalization coefficients. While low coefficient features are interpreted detrimental to the learning process since they might induce error variance into the data, high coefficient features are interpreted as important information to solve learning problem. A frequently used feature selection approach is based on the Lasso algorithm, as the Lasso implements an L1 regularization norm that penalizes non-sparsity of the model and thus pushes unnecessary model weights to zero. The Lasso coefficients can then be used to select the most important features.

**Setup.** The Lasso implementation is imported from scikit-learn, and in order to prepare it as a feature selection tool, accessed via a simple wrapper class provided in PHOTONAI. The wrapper sorts the fitted model's coefficients and only features falling in the top k percentile are kept. Both the Lasso's alpha parameter as well as the percentile of features to keep can be optimized. We add the pipeline element *LassoFeatureSelection* as given in Listing 6 between the *SimpleImputer* pipeline element and the estimator switch. Again, we run the analysis and evaluate only the validation set (*Hyperpipe* parameter *use_test_set* is set to *False*).

**Results.** The performance metrics for the pipeline with Lasso Feature Selection are given in Table 1. We see a minor performance decrease of approximately 1%. Apparently, linear feature selection is unhelpful indicating that the learning problem is rather under- than over-described by the features given. Interestingly, while 90% of the subjects are correctly identified as survivors (specificity of 91%), a notable amount of actual deaths are missed (sensitivity of 73%). The lower sensitivity in relation to a high specificity might be due to the class imbalance present in the data with more subjects surviving than dying (68%), which we will now investigate in use case 3.

**Listing 7**. Code for adding class balancing algorithms to the pipeline and optimizing the concrete class balancing strategy.

```
1 pipe += PipelineElement('ImbalancedDataTransformer',
```

```
2                                    hyperparameters={'method_name':
                                                     ['RandomUnderSampler',
3                                                    'RandomOverSampler',
4                                                    'SMOTE']})
```

## Use case 3—Handling class imbalance

As a next step, we will try to enhance predictive accuracy and balance the trade-off between specificity and sensitivity by decreasing class imbalance.

**Setup.**   In order to conveniently access class balancing algorithms, PHOTONAI offers a wrapper calling over- and under-sampling (or a combination of both) techniques implemented in the imbalanced-learn package. We remove the *LassoFeatureSelection* pipeline element and substitute it with an *ImbalancedDataTransformer* pipeline element as shown in Listing 7. As a hyperparameter, we optimize the specific class balancing method itself by evaluating random undersampling, random oversampling, and a combination of both called SMOTE.

**Results.**   Rerunning the analysis with a class balancing algorithm yields a slightly better performance (f1 score = 0.76, Matthews correlation coefficient = 0.66, see line 3 in Table 1). The optimal class balancing method was found to be SMOTE, a combination of under- and over-sampling. More importantly, a greater balance between sensitivity (82%) and specificity (86%) was reached which also resulted in a higher balanced accuracy compared to the two previous pipelines (BACC = 84%).

## Use case 4—Estimating final model performance

From the results of use cases 2 and 3, we can see that only class balancing but not feature selection slightly increased the classification performance in this specific dataset. Additionally, when we examine the results of the three learning algorithms of the class balancing pipeline, we can further see that the Random Forest (f1 = 0.76) is still outperforming gradient boosting and the Support Vector Machine (see Table 2). Therefore, we restrict our final machine learning pipeline to a class balancing element and a Random Forest classifier.

**Listing 8**. Changes made to the PHOTONAI script to generate the final model

```
1 pipe = Hyperpipe(...
2             use_test_set = True,
3             optimizer='grid_search',
4             optimizer_params = {},
5             performance_constraints = MinimumPerformanceConstraint
                                              ('f1_score',
6                                             threshold = 0.7,
7                                             strategy="mean")
8             ...)
```

**Table 2. Different estimator's average best validation performance for the class balancing pipeline.**

| Estimator | f1 | matthews corr | BACC | sens | spec |
|---|---|---|---|---|---|
| Random Forest | 0.7623 | 0.6602 | 0.8368 | 0.8161 | 0.8575 |
| Gradient Boosting | 0.7393 | 0.6233 | 0.8192 | 0.7949 | 0.8435 |
| SVM | 0.7017 | 0.5717 | 0.7895 | 0.7445 | 0.8344 |

*Notes*: matthews corr = Matthews correlation coefficient, BACC = balanced accuracy, sens = sensitivity, spec = specificity

```
9 ...
10 pipe += PipelineElement('RandomForestClassifier',
11                 criterion='gini',
12                 bootstrap = True,
13                 hyperparameters={'min_samples_split':
                                        IntegerRange(2, 30),
14                         'max_features': ['auto', 'sqrt',
                                        'log2']})
15 ...
16 pipe.fit(X,y)
```

**Setup.**  In this last step, we finish model development and estimate the final model performance. We remove the estimator switch from the pipeline and substitute it by a single Random Forest pipeline element. In addition, we decide to thoroughly investigate the hyperparameter space and therefore change the hyperparameter optimizer to grid search (see line 3-4 in Listing 8). In addition, we use the previously calculated validation metrics as a rough guide to specify a lower performance bound that promising hyperparameter configurations must outperform. Specifically, we apply a *MinimumPerformanceConstraint* on f1 score, meaning that inner fold calculations are aborted when the mean performance is below 0.7 (see line 5-7 in Listing 8). Thereby, less promising configurations are dismissed early and computational resources are saved. Importantly, we will now set *use_test_set* to *True* to make sure that PHOTONAI will evaluate the best hyperparameter configurations on the outer cv test set.

**Results.**  The final model performance on the test set is given in Table 3. All metrics remained stable when being evaluated on the previously unused test set. As a comparison, Chicco et al. (2020) [30] trained several learning algorithms to the heart failure dataset used in this example (see row 2 of table 11 in Chicco et al. [30]). PHOTONAI is able to outperform the best model of Chicco et al. which was trained on all available features in a similar fashion (see Table 3). For the f1 score, the PHOTONAI pipeline reaches 0.746. Also, sensitivity and specificity appears to be more balanced in comparison to Chicco et al.

Fig 4 shows a parallel plot of the hyperparameter space PHOTONAI has explored in this final analysis. Since we have used a grid search optimizer, all possible hyperparameter combinations have been evaluated. Interestingly, when looking at Fig 4, a clear disadvantage becomes evident when no class balancing algorithm is used, random under-sampling appears to provide generally better model performance.

## Discussion

We introduced PHOTONAI, a high-level Python API for rapid machine learning model development. As demonstrated in the example above, both the pipeline and the training and test procedure, as well the integration of hyperparameter optimization can be implemented in a few lines of code. In addition, experimenting with different algorithm sequences, hyperparameter optimization strategies, and other workflow-related parameters was realized by adding single lines of code or changing a few keywords. Through the automation of the training, validation and test procedure, data transformation and feature selection steps are restricted to the

**Table 3. Test performance metrics for the final model.**

|  | *f1* | *matthews corr* | *BACC* | *sens* | *spec* |
|---|---|---|---|---|---|
| Chicco et al. | 0.714 | 0.607 | 0.818 | 0.780 | 0.856 |
| Final PHOTONAI model | 0.746 | 0.619 | 0.818 | 0.813 | 0.823 |

*Notes*: matthews corr = Matthews correlation coefficient, BACC = balanced accuracy, sens = sensitivity, spec = specificity

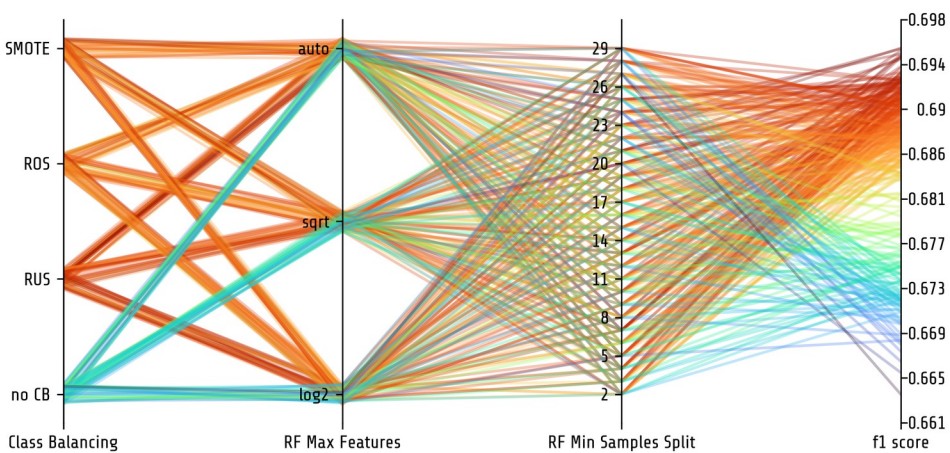

**Fig 4. Parallel plot showing hyperparameter exploration.** All three hyperparameters used in the final model are shown on the x-axis. They include the class balancing algorithm and two hyperparameters of the Random Forest classifier (maximum number of features and minimum samples per split). Each line represents a specific combination of all hyperparameters. The line color reflects the corresponding model performance based on the f1 score. Higher model performance is shown in dark red while lower model performance is shown in blue. Random under-sampling appears to increase model performance slightly while using no class balancing algorithm decreases overall model performance. CB = class balancing, RUS = random under-sampling, ROS = random over-sampling, SMOTE = synthetic minority oversampling technique, RF = Random Forest.

validation set only, thus strictly avoiding data leakage even when used by non-experts. Interestingly, this fundamentally important separation between training and test data was apparently not implemented for the feature selection analyses in Chicco et al., as they seem to have selected the most important features on the whole dataset which has most likely inflated their final model performance. Examples like this again highlight the importance of easy-to-use nested cross-validation frameworks that guarantee an unbiased estimate of the predictive performance and generalization error, which is key to, e.g., the development of reliable machine learning applications in the medical domain. Finally, the toolbox automatically identified the best hyperparameter configurations, yielded in-depth information about both validation and test set performance, and offered convenient estimator comparison tools.

PHOTONAI is developed with common scientific use cases in mind, for which it can significantly decrease programmatic overhead and support rapid model prototyping. However, use cases that substantially differ in the amount of available data or in the computational resources required to train the model, might require a different model development workflow. For example, while all kinds of neural networks can be integrated in PHOTONAI, developing and optimizing extremely complex and specialized deep neural networks with specialized architecture optimization protocols might be cumbersome within the PHOTONAI framework. In addition, unbiased performance evaluation in massive amounts of data might significantly relax the need for strict cross-validation schemes.

In addition, cross-toolbox access to algorithms comes at the cost of manually pre-registrating the algorithms with the PHOTONAI Registry system. In addition, if the algorithm does not inherently adheres to the scikit-learn object API, the user needs to manually write a wrapper class calling the algorithm according to the fit-predict-transform interface. However, this process is only required once and can afterwards be shared with the community, thereby enabling convenient access for other researchers without further effort. Furthermore, once registered, all integrated algorithms are instantaneously compatible with all other

functionalities of the PHOTONAI framework, for example, can they be optimized with any hyperparameter optimization algorithm of choice.

In the future, we intend to extend both functionality and usability. First, we will incorporate additional hyperparameter optimization strategies. While this area has seen tremendous progress in recent years, these algorithms are often not readily available to data scientists, and studies systematically comparing them are extremely scarce. Second, we seek to extend automatic ensemble generation to fully exploit the various models trained during the hyperparameter optimization process. Generally, we strive to pre-register more of the arising ML utility packages, so that accessibility is facilitated and functionality can be used within PHOTONAI as a unified framework. Finally, we would like to improve our convenience functions for model performance assessment and visualization.

In addition to these core functionalities, we aim to establish an ecosystem of add-on modules simplifying ML analyses for different data types and modalities. For example, we will add a neuroimaging module as a means to directly use multimodal Magnetic Resonance Imaging (MRI) data in ML analyses. In addition, a graph module will integrate existing graph analysis functions and provide specialized ML approaches for graph data. Likewise, modules integrating additional data modalities such as omics data would be of great value. More generally, PHOTONAI would benefit from modules making novel approaches to model interpretation (i.e. Explainability) available.

## Conclusion

In summary, PHOTONAI is especially well-suited in contexts requiring rapid and iterative evaluation of novel approaches such as applied ML research in medicine and the Life Sciences. In the future, we hope to attract more developers and users to establish a thriving, open-source community.

## Author Contributions

**Conceptualization:** Ramona Leenings, Nils Ralf Winter, Tim Hahn.

**Formal analysis:** Ramona Leenings, Benjamin Risse, Xiaoyi Jiang, Tim Hahn.

**Funding acquisition:** Udo Dannlowski, Tim Hahn.

**Project administration:** Ramona Leenings, Nils Ralf Winter.

**Software:** Ramona Leenings, Nils Ralf Winter, Lucas Plagwitz, Vincent Holstein, Jan Ernsting, Kelvin Sarink, Lukas Fisch, Jakob Steenweg, Leon Kleine-Vennekate, Julian Gebker, Daniel Emden, Dominik Grotegerd, Tim Hahn.

**Supervision:** Tim Hahn.

**Writing – original draft:** Ramona Leenings, Nils Ralf Winter, Tim Hahn.

**Writing – review & editing:** Ramona Leenings, Nils Ralf Winter, Vincent Holstein, Dominik Grotegerd, Nils Opel, Benjamin Risse, Xiaoyi Jiang, Udo Dannlowski, Tim Hahn.

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
