## [Decision Letter · Decision Letter 0]

19 Apr 2021

PONE-D-21-10500

PHOTONAI - A Python API for rapid machine learning model development

PLOS ONE

Dear Dr. Leenings,

Thank you for submitting your manuscript to PLOS ONE. After careful consideration, we feel that it has merit but does not fully meet PLOS ONE’s publication criteria as it currently stands. Therefore, we invite you to submit a revised version of the manuscript that addresses the points raised during the review process.

Based on the comments received form the reviewers and my own observation, I recommend major revisions for the paper.

We look forward to receiving your revised manuscript.

Kind regards,

Thippa Reddy Gadekallu

Academic Editor

PLOS ONE

Journal Requirements:

Reviewers' comments:

Reviewer's Responses to Questions

**Comments to the Author**

1. Is the manuscript technically sound, and do the data support the conclusions?

Reviewer #1: Yes

Reviewer #2: Yes

Reviewer #3: Yes

2. Has the statistical analysis been performed appropriately and rigorously? 

Reviewer #1: N/A

Reviewer #2: Yes

Reviewer #3: Yes

3. Have the authors made all data underlying the findings in their manuscript fully available?

Reviewer #1: Yes

Reviewer #2: Yes

Reviewer #3: Yes

4. Is the manuscript presented in an intelligible fashion and written in standard English?

Reviewer #1: Yes

Reviewer #2: Yes

Reviewer #3: Yes

5. Review Comments to the Author

Reviewer #1: Authors have developed a high-level Python API to simplify and accelerate machine learning model development.

Abstract looks promising, but can be improved to summarize the work in a more detailed way.

The developed software is well-suited for different research applications because project specific custom code can be integrated easily within the machine learning pipeline.

Introduction should be revised: (1) Introduce the problem (2)discuss about some of the existing solutions (3)identify the gap or scope of improvement (4) discuss in order to address the identified gaps what is the methodology used (5) list out the contributions.

Related work in the same field should be written in detail along with the limitations.

Authors may refer to the following articles:

Towards Secure Data Fusion in Industrial IoT using Transfer Learning

Applications in Security and Evasions in Machine Learning: A Survey

Long-term Wind Power Forecasting using Tree-based Learning Algorithms

Algorithm and Framework can be explained in a detailed and organized way.

Some paragraphs in the paper lack connectivity.

Future work should be mentioned after mentioning limitations of PHOTONAI.

There are many grammatical errors along with typo.

Reviewer #2: Authors of this paper presents a high-level abstract of newly developed Python API named PHOTONAI which simplify and accelerate the machine learning model development. Very interesting and high level technical paper. The novelty was clearly specified and explained. Very good use of language and very well written. The demonstration is a key for this paper. A high level of demonstration with case scenarios would have added more value to the paper. Literature review of the paper can be extended. Overall, a very good paper.

Reviewer #3: Some of the comments to improve the quality.

1. Abstract add the results achieved

2. In introduction add the contributions

3. Add the latest references

4. In results make more tables and figures to justify your work.

5. The authors can cite the following references

(An AI-based intelligent system for healthcare analysis using Ridge-Adaline Stochastic Gradient Descent Classifier,

Genetically Optimized Prediction of Remaining Useful Life)

6. PLOS authors have the option to publish the peer review history of their article (what does this mean?). If published, this will include your full peer review and any attached files.

Reviewer #1: **Yes: **Rutvij H Jhaveri

Reviewer #2: No

Reviewer #3: **Yes: **Praveen Kumar Reddy Maddikunta

---

## [Author Response · Author response to Decision Letter 0]

7 Jun 2021

REVIEWER #1 

Authors have developed a high-level Python API to simplify and accelerate machine learning model development.

Abstract looks promising, but can be improved to summarize the work in a more detailed way.

We thank Reviewer 1 for this helpful suggestion and have added further details to the abstract

p. 1: “[...] It functions as a unifying framework allowing the user to easily access and combine algorithms from different toolboxes into custom algorithm sequences. It is especially designed to support the iterative model development process and automates the repetitive training, hyperparameter optimization and evaluation tasks. Importantly, the workflow ensures unbiased performance estimates while still allowing the user to fully customize the machine learning analysis. [...] Its practical utility is demonstrated on an exemplary medical machine learning problem, achieving a state-of-the-art solution in few lines of code. [...]”

The developed software is well-suited for different research applications because project specific custom code can be integrated easily within the machine learning pipeline.

Introduction should be revised: (1) Introduce the problem (2)discuss about some of the existing solutions (3)identify the gap or scope of improvement (4) discuss in order to address the identified gaps what is the methodology used (5) list out the contributions.

Related work in the same field should be written in detail along with the limitations.

Authors may refer to the following articles:

Towards Secure Data Fusion in Industrial IoT using Transfer Learning

Applications in Security and Evasions in Machine Learning: A Survey

Long-term Wind Power Forecasting using Tree-based Learning Algorithms

We thank Reviewer 1 for this concrete feedback and have restructured the introduction to improve the line of reasoning and integrate the suggested structure. Specifically, in the second section of the introduction (now called ‘Existing solutions’), we have revised the description of existing software packages and have added more detail on the distinction between these packages and the proposed PHOTONAI software. We have also added additional headings to make the separation between these paragraphs clearer, which should make it easier for the reader to follow the argument. Additionally, we have added a completely new section on current shortcomings of existing solutions to further emphasize the contributions we aim to provide with PHOTONAI.

p. 2: (1) Introduce the problem:

“In recent years, the interest in machine learning for medical, biological, and life science research has significantly increased. Technological advances develop with breathtaking speed. The basic workflow to construct, optimize and evaluate a machine learning model, however, has remained virtually unchanged. In essence, it can be framed as the (systematic) search for the best combination of data processing steps, learning algorithms, and hyperparameter values under the premise of unbiased performance estimation.

Subject to the iteratively optimized workflow is a machine learning pipeline, which in this context is defined as the sequence of algorithms subsequently applied to the data. To begin with, the data is commonly prepared by successively applying several processing steps such as normalization, imputation, feature selection, dimensionality reduction, data augmentation, and others. The altered data is then forwarded to one or more learning algorithms which internally derive the best fit for the learning task and finally yield predictions.

In practice, researchers select suitable preprocessing and learning algorithms from different toolboxes, learn toolbox-specific syntaxes, decide for a training and testing scheme, manage the data flow and, over time, iteratively optimize their choices. Importantly, all of this is done while preventing data leakage, calculating performance metrics, adhering to (nested) cross-validation best practices, and searching for the optimal (hyperparameter-) configuration.

A multitude of high-quality and well-maintained open-source toolboxes offer specialized solutions, each for a particular subdomain of machine learning-related (optimization) problems.”

p. 3-4: (2) Discuss existing solutions. 

“In the field of (deep) neural networks for example, libraries such as Tensorflow, Theano, Caffe and PyTorch offer domain-specific implementations for nodes, layers, optimizers, as well as evaluation and utility functions. On top of that, higher level Application Programming Interfaces (APIs) such as Keras and fastai, offer expressive syntaxes for accelerated and enhanced development of deep neural network architectures. 

In the same manner, the scikit-learn toolbox has evolved as one of the major resources of the field, covering a very broad range of regression-, classification-, clustering- and preprocessing algorithm implementations. It has established the de-facto standard interface for data-processing and learning algorithms, and, in addition, offers a wide range of utility functions, such as cross-validation schemes and model evaluation metrics.

Next to these general frameworks, other libraries in the software landscape offer functionalities to address more specialized problems. Prominent examples are the imbalanced-learn toolbox, which provides numerous over- or under- sampling methods, or modality-specific libraries such as nilearn and nibabel which offer utility functions for accessing and preparing neuroimaging data.

On top of that, the software landscape is complemented by several hyperparameter optimization packages, each implementing a different strategy to find the most effective hyperparameter combination. Next to Bayesian approaches, such as Scikit-optimize or SMAC, there are packages implementing evolutionary strategies or packages approximating gradient descent within the hyperparameter space. Each package requires specific syntax and unique hyperparameter space definitions.

Finally, there are approaches uniting all these components into algorithms that automatically derive the best model architecture and hyperparameter settings for a given dataset. Libraries such as auto-sklearn, TPOT, AutoWeka, Auto-keras, AutoML, Auto-Gluon and others optimize a specific set of data-processing methods, learning algorithms and their respective hyperparameters. While very intriguing, these libraries aim at full automation - neglecting the need for customization and foregoing the opportunity to incorporate high-level domain knowledge in the model architecture search. Especially the complex and often high-dimensional data structure native to medical and biological research requires the integration and application of modality-specific processing and often entails the development of novel algorithms.”

p.5: (3) Identify the gap or scope of improvement

“Currently, iterative model development approaches across different toolboxes as well as design and optimization of custom algorithm sequences are barely supported. For a start, scikit-learn has introduced the concept of pipelines, which successively apply a list of processing methods (referred to as transformers) and a final learning algorithm (called estimator) to the data. The pipeline directs the data from one algorithm to another and can be trained and evaluated in (simple) cross-validation schemes, thereby significantly reducing programmatic overhead. Scikit-learn's consistent usage of standard interfaces enables the pipeline to be subject to scikit-learn's inherent hyperparameter optimization strategies based on random- and grid-search. While being a simple and effective tool, there still are several limitations. For one, hyperparameter optimization requires a nested cross-validation scheme, which is not inherently enforced. Second, a standardized solution for easy integration of custom or third-party algorithms is not considered. In addition, several repetitive tasks, such as metric calculations, logging, and visualization lack automation and still need to be handled manually. Finally, the pipeline can not handle adjustments to the target vector, thereby excluding algorithms for e.g. data augmentation or handling class imbalance.”

p. 5: (4) Discuss methodology to address identified issues

“To address these issues, we propose PHOTONAI as a high-level Python API that acts as a mediator between different toolboxes. Established solutions are conveniently accessible or can be easily added. It combines an automated supervised machine learning workflow with the concept of custom machine learning pipelines. Thereby it is able to considerably accelerate design iterations and simplify the evaluation of novel analysis pipelines.”

p. 5-6: (5) List out contributions

Increased Accessibility. By pre-registering data processing methods, learning algorithms, hyperparameter optimization strategies, performance metrics, and other functionalities, the user can effortlessly access established machine learning implementations via simple keywords. In addition, by relying on the established scikit-learn object API, users can easily integrate any third-party or custom algorithm implementation.

Extended Pipeline Functionality. A simple to use class structure allows the user to arrange selected algorithms into single or parallel pipeline sequences. Extending the pipeline concept of scikit-learn, we add novel functionality such as flexible positioning of learning algorithms, target vector manipulations, callback functions, specialized caching, parallel data-streams, Or-Operations, and other features as described below.

Automation. PHOTONAI can automatically train, (hyperparameter-) optimize and evaluate any custom pipeline. Importantly, the user designs the training and testing procedure by selecting (nested) cross-validation schemes, hyperparameter optimization strategies, and performance metrics from a range of pre-integrated or custom-built options. Thereby, development time is significantly decreased and conceptual errors such as information leakage between training, validation, and test set are minimized. Training information, baseline performances, hyperparameter optimization progress, and test performance evaluations are persisted and can be visualized via an interactive, browser-based graphical interface (PHOTONAI Explorer) to facilitate model improvement.

Model Sharing. A standardized format for saving, loading, and distributing optimized and trained pipeline architectures enables model sharing and external model validation even for non-expert users.”

Algorithm and Framework can be explained in a detailed and organized way.

Some paragraphs in the paper lack connectivity.

We thank reviewer #1 for outlining this shortcoming and have updated the section introducing the algorithm and the framework, as well as the description of the software’s features. To further improve the readability of these sections, we have added additional headings to the respective paragraphs.

p. 6-7: “In the following, we will describe the automated supervised machine learning workflow implemented in PHOTONAI. Subsequently, we will outline the class structure, which is the core of its expressive syntax. At the same time, we will highlight its current functionalities, and finally, provide a hands-on example to introduce PHOTONAI's usage.

PHOTONAI automatizes the supervised machine learning workflow according to user-defined parameters (see pseudocode in Listing~1). In a nutshell, cross-validation folds are derived to iteratively train and evaluate a machine learning pipeline following the hyperparameter optimization strategy's current parameter value suggestions. Performance metrics are calculated, the progress is logged and finally, the best hyperparameter configuration is selected to train a final model. The training, testing, and optimization workflow is automated, however, it is important to note that it is parameterized by user choices and therefore fully customized.”

p. 7: “In order to achieve an efficient and expressive customization syntax, PHOTONAI's class architecture captures all workflow- and pipeline-related parameters into distinct and combinable components (see Fig~1). A central management class called Hyperpipe - short for hyperparameter optimization pipeline - handles the setup of the pipeline and executes the training and test procedure according to user choices. Basis to the data flow is a custom Pipeline implementation, which streams data through a sequence of PipelineElement objects, the latter of which represent either established or custom algorithm implementations. In addition, clear interfaces and several utility classes allow the integration of custom solutions, adjust the training and test procedure and build parallel data streams. In the following, PHOTONAI's core classes and their respective features will be further detailed. 

PHOTONAI's core functionality is encapsulated in a class called Hyperpipe, which controls all workflow and pipeline-related parameters and manages the cross-validated training and testing procedure. [...]”

p.9: “The Hyperpipe relies on a custom pipeline implementation that is conceptually related to the scikit-learn pipeline but extends it with four core features. [...]”

p.10: “In order to integrate a particular algorithm into the pipeline's data stream, PHOTONAI implements the PipelineElement class. [...]”

Future work should be mentioned after mentioning limitations of PHOTONAI.

We agree with reviewer #1 and have introduced a discussion of limitations before mentioning future work. 

p. 23: “We introduced PHOTONAI, a high-level Python API for rapid machine learning model development. As demonstrated in the example above, both the pipeline and the training and test procedure, as well the integration of hyperparameter optimization can be implemented in a few lines of code. In addition, experimenting with different algorithm sequences, hyperparameter optimization strategies, and other workflow-related parameters was realized by adding single lines of code or changing a few keywords. Through the automation of the training, validation and test procedure, data transformation and feature selection steps are restricted to the validation set only, thus strictly avoiding data leakage even when used by non-experts. Interestingly, this fundamentally important separation between training and test data was apparently not implemented for the feature selection analyses in Chicco et al., as they seem to have selected the most important features on the whole dataset which has most likely inflated their final model performance. Examples like this again highlight the importance of easy-to-use nested cross-validation frameworks that guarantee an unbiased estimate of the predictive performance and generalization error, which is key to, e.g., the development of reliable machine learning applications in the medical domain. Finally, the toolbox automatically identified the best hyperparameter configurations, yielded in-depth information about both validation and test set performance, and offered convenient estimator comparison tools.

PHOTONAI is developed with common scientific use cases in mind, for which it can significantly decrease programmatic overhead and support rapid model prototyping. However, use cases that substantially differ in the amount of available data or in the computational resources required to train the model, might require a different model development workflow. For example, while all kinds of neural networks can be integrated in PHOTONAI, developing and optimizing extremely complex and specialized deep neural networks with specialized architecture optimization protocols might be cumbersome within the PHOTONAI framework. In addition, unbiased performance evaluation in massive amounts of data might significantly relax the need for strict cross-validation schemes.

In addition, cross-toolbox access to algorithms comes at the cost of manually pre-registrating the algorithms with the PHOTONAI Registry system. In addition, if the algorithm does not inherently adheres to the scikit-learn object API, the user needs to manually write a wrapper class calling the algorithm according to the fit-predict-transform interface. However, this process is only required once and can afterwards be shared with the community, thereby enabling convenient access for other researchers without further effort. Furthermore, once registered, all integrated algorithms are instantaneously compatible with all other functionalities of the PHOTONAI framework, for example, can they be optimized with any hyperparameter optimization algorithm of choice. 

In the future, we intend to extend both functionality and usability. [...]“

There are many grammatical errors along with typo.

We apologize for this shortcoming and have double-checked typos and grammatical errors in the entire manuscript. 

REVIEWER #2

Authors of this paper presents a high-level abstract of newly developed Python API named PHOTONAI which simplify and accelerate the machine learning model development. Very interesting and high level technical paper. The novelty was clearly specified and explained. Very good use of language and very well written. The demonstration is a key for this paper. [...] Overall, a very good paper.

We thank reviewer#2 for his expertise and the overall good evaluation of the paper.

A high level of demonstration with case scenarios would have added more value to the paper. 

We agree with reviewer#2 and have divided the example section into four separate use cases that highlight different aspects of the toolbox’s functionalities. We have also adjusted the syntax highlighting and formatting of the example Python code to make it easier to read and follow along.

p. 14-22: 

Hyperpipe Setup

First, we will define the training, optimization, and evaluation workflow in an initial call to the Hyperpipe class. The python code in Listing 4 shows the PHOTONAI code defining both the data flow and the training and test procedure. After importing the relevant packages and loading the data, we instantiate a Hyperpipe and choose the workflow parameters as follows:

For the outer cross-validation loop, we specify 100 shuffled iterations each holding out a test set of 20 percent. For the inner cross-validation loop, we select a ten-fold cross-validation. (lines 13-14).

To measure model performance, we specify that f1 score, Matthews correlation coefficient, balanced accuracy, as well as sensitivity and specificity are to be calculated (lines 16-17).

We optimize the pipeline for f1 score, as it maximizes both sensitive and specific predictions, which is particularly important in medical contexts. (line 18).

To save computational resources and time, we enable caching by specifying a cache folder (line 22). This is particularly useful in examples where there are a lot of partially overlapping hyperparameters to be tested.

Finally, we specify a folder to which the output is written (line 21) and set the verbosity of the console log to 1. At this verbosity level, information on every tested hyperparameter configuration and its respective performance estimate is printed to the console.

After the hyperpipe has been defined, we can design the flow of the data by adding algorithms and respective hyperparameters to the pipeline.

First, data is normalized using scikit-learn's StandardScaler which both centers the data and scales it to unit variance (line 26).

Second, we impute missing values with the mean values per feature of the training set by calling scikit-learn's SimpleImputer (line 27). 

Of note, we consider use cases 1 to 3 (see below) to be exploratory analyses. We believe this simulates a naturalistic workflow of machine learning projects where different algorithms, feature preprocessing and hyperparameters are tested in a manual fashion. However, if done incorrectly, this inevitably leads to a manual over-fitting to the data at hand, which is especially troublesome in high-stake medical problems with small datasets. In this context, manual over-fitting happens implicitly when data scientists optimize algorithms and hyperparameters by repeatedly looking at cross-validated test performance. In PHOTONAI, this problem can easily be avoided by setting the Hyperpipe parameter use_test_set to False. This way, PHOTONAI will still apply nested cv but will only report validation performances from the inner cv loop, not the outer cv test data. In the final use case 4, use_test_set is set to True to estimate final model performance and generalizability on the actual test sets.

Use case 1 - Estimator selection

Although some rules of thumb for selecting the correct algorithm do exist, knowing the optimal learning algorithm for a specific task a priori is impossible (no free lunch theorems). Therefore, the possibility to automatically evaluate multiple algorithms within nested cross-validation is crucial to efficient and unbiased machine learning analyses. In this example, we first train a machine learning pipeline and consider three different learning algorithms that we find appropriate for this learning problem. These algorithms are added to the PHOTONAI Hyperpipe in addition to the scaling and imputing preprocessing steps defined above.

Setup

To compare different learning algorithms, an Or-Element called Switch is added to the pipeline that toggles between several learning algorithms (see Listing 5). Here, we compare a random forest (RF), gradient boosting (GB), and a support vector machine (SVM) against each other. Again, all algorithms are imported from scikit-learn, and for every element we specify algorithm-specific hyperparameters that are automatically optimized.

To efficiently optimize hyperparameters of different learning algorithms, the switch optimizer in PHOTONAI can be used which optimizes each learning algorithm in an individual hyperparameter space (line 19 in Listing). We apply Bayesian optimization to each space respectively and limit the number of tested configurations to 10 (line 20 in Listing 4).

Finally, we can start the training, optimization, and test procedure by calling Hyperpipe.fit(). After the pipeline optimization has finished, we extract not only the overall best hyperparameter configuration and its respective performance, but also the best configuration performance per learning algorithm (RF, GB, SVM, see line 21 in Listing).

Results

The results of the initial estimator selection analysis are given in the first line of Table 1. We observe an f1 score of 75% and a Matthews correlation coefficient of 65%. The best config found by the hyperparameter optimization strategy applied the Random Forest classifier, which thus in this case outperforms gradient boosting and the Support Vector Machine.

Use case 2 - Feature selection

Next, we will evaluate the effect of an additional feature selection step. This can be done, e.g., by analyzing a linear model's normalization coefficients. While low coefficient features are interpreted detrimental to the learning process since they might induce error variance into the data, high coefficient features are interpreted as important information to solve learning problem. A frequently used feature selection approach is based on the Lasso algorithm, as the Lasso implements an L1 regularization norm that penalizes non-sparsity of the model and thus pushes unnecessary model weights to zero. The Lasso coefficients can then be used to select the most important features.

Setup

The Lasso implementation is imported from scikit-learn, and in order to prepare it as a feature selection tool, accessed via a simple wrapper class provided in PHOTONAI. The wrapper sorts the fitted model's coefficients and only features falling in the top k percentile are kept. Both the Lasso's alpha parameter as well as the percentile of features to keep can be optimized. We add the pipeline element LassoFeatureSelection as given in Listing 6 between the SimpleImputer pipeline element and the estimator switch. Again, we run the analysis and evaluate only the validation set (Hyperpipe parameter use_test_set is set to False).

Results

The performance metrics for the pipeline with Lasso Feature Selection are given in Table 1. We see a minor performance decrease of approximately 1%. Apparently, linear feature selection is unhelpful indicating that the learning problem is rather under- than over-described by the features given. Interestingly, while 90% of the subjects are correctly identified as survivors (specificity of 91%), a notable amount of actual deaths are missed (sensitivity of 73%). The lower sensitivity in relation to a high specificity might be due to the class imbalance present in the data with more subjects surviving than dying (68%), which we will now investigate in use case 3.

Use case 3 - Handling class imbalance

As a next step, we will try to enhance predictive accuracy and balance the trade-off between specificity and sensitivity by decreasing class imbalance.

Setup

In order to conveniently access class balancing algorithms, PHOTONAI offers a wrapper calling over- and under-sampling (or a combination of both) techniques implemented in the imbalanced-learn package. We remove the LassoFeatureSelection pipeline element and substitute it with an ImbalancedDataTransformer pipeline element as shown in Listing 7. As a hyperparameter, we optimize the specific class balancing method itself by evaluating random undersampling, random oversampling, and a combination of both called SMOTE.

Results

Rerunning the analysis with a class balancing algorithm yields a slightly better performance (f1 score = 0.76, Matthews correlation coefficient = 0.66, see line 3 in Table 1). The optimal class balancing method was found to be SMOTE, a combination of under- and over-sampling. More importantly, a greater balance between sensitivity (82%) and specificity (86%) was reached which also resulted in a higher balanced accuracy compared to the two previous pipelines (BACC = 84%).

Use case 4 - Estimating final model performance

From the results of use cases 2 and 3, we can see that only class balancing but not feature selection slightly increased the classification performance in this specific dataset. Additionally, when we examine the results of the three learning algorithms of the class balancing pipeline, we can further see that the Random Forest (f1 = 0.76) is still outperforming gradient boosting and the Support Vector Machine (see Table 2). Therefore, we restrict our final machine learning pipeline to a class balancing element and a Random Forest classifier.

Setup

In this last step, we finish model development and estimate the final model performance. We remove the estimator switch from the pipeline and substitute it by a single Random Forest pipeline element. In addition, we decide to thoroughly investigate the hyperparameter space and therefore change the hyperparameter optimizer to grid search (see line 3-4 in Listing 8). In addition, we use the previously calculated validation metrics as a rough guide to specify a lower performance bound that promising hyperparameter configurations must outperform. Specifically, we apply a MinimumPerformanceConstraint on f1 score, meaning that inner fold calculations are aborted when the mean performance is below 0.7 (see line 5-7 in Listing 8). Thereby, less promising configurations are dismissed early and computational resources are saved. Importantly, we will now set use_test_set to True to make sure that PHOTONAI will evaluate the best hyperparameter configurations on the outer cv test set.

Results

The final model performance on the test set is given in Table 3. All metrics remained stable when being evaluated on the previously unused test set. As a comparison, Chicco et al. (2020) trained several learning algorithms to the heart failure dataset used in this example (see row 2 of Table 11 in Chicco et al.). PHOTONAI is able to outperform the best model of Chicco et al. which was trained on all available features in a similar fashion (see Table 3). For the f1 score, the PHOTONAI pipeline reaches 0.746. Also, sensitivity and specificity appears to be more balanced in comparison to Chicco et al. 

Fig 4 shows a parallel plot of the hyperparameter space PHOTONAI has explored in this final analysis. Since we have used a grid search optimizer, all possible hyperparameter combinations have been evaluated. Interestingly, when looking at Figure 4, a clear disadvantage becomes evident when no class balancing algorithm is used, random under-sampling appears to provide generally better model performance. 

Literature review of the paper can be extended. 

We thank reviewer #2 for this useful comment and have updated the related section.

p. 3: “A multitude of high-quality and well-maintained open-source toolboxes offer specialized solutions, each for a particular subdomain of machine learning-related (optimization) problems.

In the field of (deep) neural networks for example, libraries such as Tensorflow, Theano, Caffe and PyTorch offer domain-specific implementations for nodes, layers, optimizers, as well as evaluation and utility functions. On top of that, higher level Application Programming Interfaces (APIs) such as Keras and fastai, offer expressive syntaxes for accelerated and enhanced development of deep neural network architectures. 

In the same manner, the scikit-learn toolbox has evolved as one of the major resources of the field, covering a very broad range of regression-, classification-, clustering- and preprocessing algorithm implementations. It has established the de-facto standard interface for data-processing and learning algorithms, and, in addition, offers a wide range of utility functions, such as cross-validation schemes and model evaluation metrics.

Next to these general frameworks, other libraries in the software landscape offer functionalities to address more specialized problems. Prominent examples are the imbalanced-learn toolbox,which provides numerous over- or under- sampling methods, or modality-specific libraries such as nilearn and nibabel which offer utility functions for accessing and preparing neuroimaging data.

On top of that, the software landscape is complemented by several hyperparameter optimization packages, each implementing a different strategy to find the most effective hyperparameter combination. Next to Bayesian approaches, such as Scikit-optimize or SMAC, there are packages implementing evolutionary strategies or packages approximating gradient descent within the hyperparameter space. Each package requires specific syntax and unique hyperparameter space definitions.

Finally, there are approaches uniting all these components into algorithms that automatically derive the best model architecture and hyperparameter settings for a given dataset. Libraries such as auto-sklearn, TPOT, AutoWeka, Auto-keras, AutoML, Auto-Gluon and others optimize a specific set of data-processing methods, learning algorithms and their respective hyperparameters. While very intriguing, these libraries aim at full automation - neglecting the need for customization and foregoing the opportunity to incorporate high-level domain knowledge in the model architecture search. Especially the complex and often high-dimensional data structure native to medical and biological research requires the integration and application of modality-specific processing and often entails the development of novel algorithms.

Currently, iterative model development approaches across different toolboxes as well as design and optimization of custom algorithm sequences are barely supported. For a start, scikit-learn has introduced the concept of pipelines, which successively apply a list of processing methods (referred to as transformers) and a final learning algorithm (called estimator) to the data. The pipeline directs the data from one algorithm to another and can be trained and evaluated in (simple) cross-validation schemes, thereby significantly reducing programmatic overhead. Scikit-learn's consistent usage of standard interfaces enables the pipeline to be subject to scikit-learn's inherent hyperparameter optimization strategies based on random- and grid-search. While being a simple and effective tool, there still are several limitations. For one, hyperparameter optimization requires a nested cross-validation scheme, which is not inherently enforced. Second, a standardized solution for easy integration of custom or third-party algorithms is not considered. In addition, several repetitive tasks, such as metric calculations, logging, and visualization lack automation and still need to be handled manually. Finally, the pipeline can not handle adjustments to the target vector, thereby excluding algorithms for e.g. data augmentation or handling class imbalance.”

REVIEWER #3 

Some of the comments to improve the quality.

1. Abstract add the results achieved

We thank reviewer #3 for this helpful suggestion and have added the results to the abstract. 

p. 1: “Its practical utility is demonstrated on an exemplary medical machine learning problem for which we reach state-of-the-art performance in a few lines of code.”

2. In introduction add the contributions

We thank reviewer #3 for outlining this shortcoming and have edited the introduction to clarify the contributions. 

p. 3-4: “To address these issues, we propose PHOTONAI as a high-level Python API that acts as a mediator between different toolboxes. Established solutions are conveniently accessible or can be easily added. It combines an automated supervised machine learning workflow with the concept of custom machine learning pipelines. Thereby it is able to considerably accelerate design iterations and simplify the evaluation of novel analysis pipelines.

Increased Accessibility. By pre-registering data processing methods, learning algorithms, hyperparameter optimization strategies, performance metrics, and other functionalities, the user can effortlessly access established machine learning implementations via simple keywords. In addition, by relying on the established scikit-learn object API, users can easily integrate any third-party or custom algorithm implementation.

Extended Pipeline Functionality. A simple to use class structure allows the user to arrange selected algorithms into single or parallel pipeline sequences. Extending the pipeline concept of scikit-learn, we add novel functionality such as flexible positioning of learning algorithms, target vector manipulations, callback functions, specialized caching, parallel data-streams, Or-Operations, and other features as described below.

Automation. PHOTONAI can automatically train, (hyperparameter-) optimize and evaluate any custom pipeline. Importantly, the user designs the training and testing procedure by selecting (nested) cross-validation schemes, hyperparameter optimization strategies, and performance metrics from a range of pre-integrated or custom-built options. Thereby, development time is significantly decreased and conceptual errors such as information leakage between training, validation, and test set are minimized. Training information, baseline performances, hyperparameter optimization progress, and test performance evaluations are persisted and can be visualized via an interactive, browser-based graphical interface (PHOTONAI Explorer) to facilitate model improvement.

Model Sharing. A standardized format for saving, loading, and distributing optimized and trained pipeline architectures enables model sharing and external model validation even for non-expert users.”

3. Add the latest references

We agreed with reviewer #3 and have updated the paper, see comment 2 of Reviewer #2 

4. In results make more tables and figures to justify your work.

We thank reviewer #1 for outlining this shortcoming and have revised the complete example and result section. We have added an individual result paragraph for every use case scenario and have added the benchmark results to the final model performance table. We have also increased the readability and design of all tables. Additionally, we have added a parallel plot to visualize the explored hyperparameter space of the final analysis. 

p. 14-22: For the corresponding changes of the example and results section, please see our response to reviewer #3 (“high-level of demonstration with case scenarios”).

p.22:

Fig 4. Parallel plot showing hyperparameter exploration. All three hyperparameters used in the final model are shown on the x-axis. They include the class balancing algorithm and two hyperparameters of the Random Forest classifier (maximum number of features and minimum samples per split). Each line represents a specific combination of all hyperparameters. The line color reflects the corresponding model performance based on the f1 score. Higher model performance is shown in dark red while lower model performance is shown in blue. Random under-sampling appears to increase model performance slightly while using no class balancing algorithm decreases overall model performance. CB = class balancing, RUS = random under-sampling, ROS = random over-sampling, SMOTE = synthetic minority oversampling technique, RF = Random Forest.

5. The authors can cite the following references

(An AI-based intelligent system for healthcare analysis using Ridge-Adaline Stochastic Gradient Descent Classifier, Genetically Optimized Prediction of Remaining Useful Life)

We are happy to include the two references suggested by reviewer#3. However, due to the specificity of both papers we were unable to relate the topic and context to the current, rather general work focused on software engineering. We kindly ask the reviewer to specify where and in which context the papers could be cited in our work.

---

## [Decision Letter · Decision Letter 1]

21 Jun 2021

PHOTONAI - A Python API for rapid machine learning model development

PONE-D-21-10500R1

Dear Dr. Leenings,

We’re pleased to inform you that your manuscript has been judged scientifically suitable for publication and will be formally accepted for publication once it meets all outstanding technical requirements.

Kind regards,

Thippa Reddy Gadekallu

Academic Editor

PLOS ONE

Additional Editor Comments (optional):

Reviewers' comments:

Reviewer's Responses to Questions

**Comments to the Author**

1. If the authors have adequately addressed your comments raised in a previous round of review and you feel that this manuscript is now acceptable for publication, you may indicate that here to bypass the “Comments to the Author” section, enter your conflict of interest statement in the “Confidential to Editor” section, and submit your "Accept" recommendation.

Reviewer #2: All comments have been addressed

Reviewer #3: All comments have been addressed

2. Is the manuscript technically sound, and do the data support the conclusions?

Reviewer #2: Yes

Reviewer #3: Yes

3. Has the statistical analysis been performed appropriately and rigorously? 

Reviewer #2: Yes

Reviewer #3: Yes

4. Have the authors made all data underlying the findings in their manuscript fully available?

Reviewer #2: Yes

Reviewer #3: Yes

5. Is the manuscript presented in an intelligible fashion and written in standard English?

Reviewer #2: Yes

Reviewer #3: Yes

6. Review Comments to the Author

Reviewer #2: Authors of this manuscript did a great job on addressing the comments from the reviewers. Outstanding effort!

Reviewer #3: The authors have addressed all of my comments. The paper can can be accepted in the current format. Thank you

7. PLOS authors have the option to publish the peer review history of their article (what does this mean?). If published, this will include your full peer review and any attached files.

Reviewer #2: No

Reviewer #3: No

---

## [Editor Report · Acceptance letter]

12 Jul 2021

PONE-D-21-10500R1 

PHOTONAI - A Python API for rapid machine learning model development 

Dear Dr. Leenings:

I'm pleased to inform you that your manuscript has been deemed suitable for publication in PLOS ONE. Congratulations! Your manuscript is now with our production department. 

Kind regards, 

on behalf of

Dr. Thippa Reddy Gadekallu 

Academic Editor

PLOS ONE